# Lipidomics of Bioactive Lipids in Alzheimer’s and Parkinson’s Diseases: Where Are We?

**DOI:** 10.3390/ijms23116235

**Published:** 2022-06-02

**Authors:** Valerio Chiurchiù, Marta Tiberi, Alessandro Matteocci, Federico Fazio, Hasibullah Siffeti, Stefano Saracini, Nicola Biagio Mercuri, Giuseppe Sancesario

**Affiliations:** 1Institute of Translational Pharmacology, National Research Council, 00179 Rome, Italy; 2Laboratory of Resolution of Neuroinflammation, IRCCS Santa Lucia Foundation, 00179 Rome, Italy; m.tiberi@hsantalucia.it (M.T.); a.matteocci@hsantalucia.it (A.M.); federico_fazio@outlook.it (F.F.); siffeti@web.de (H.S.); telomerasi@hotmail.com (S.S.); 3Institute for Nutritional Medicine, University Medical Centre Schleswig Holstein, 23562 Lübeck, Germany; 4Neurology Unit, Tor Vergata Hospital, 00133 Rome, Italy; mercurin@med.uniroma2.it; 5Department of Experimental Neuroscience, IRCCS Santa Lucia Foundation, 00179 Rome, Italy; 6Department of Systems Medicine, Tor Vergata University of Rome, 00179 Rome, Italy; sancesario@med.uniroma2.it

**Keywords:** glycerophospholipids, sphingolipids, classical eicosanoids, specialized pro-resolving mediators, endocannabinoids, Alzheimer’s, Parkinson’s

## Abstract

Lipids are not only constituents of cellular membranes, but they are also key signaling mediators, thus acting as “bioactive lipids”. Among the prominent roles exerted by bioactive lipids are immune regulation, inflammation, and maintenance of homeostasis. Accumulated evidence indicates the existence of a bidirectional relationship between the immune and nervous systems, and lipids can interact particularly with the aggregation and propagation of many pathogenic proteins that are well-renowned hallmarks of several neurodegenerative disorders, including Alzheimer’s (AD) and Parkinson’s (PD) diseases. In this review, we summarize the current knowledge about the presence and quantification of the main classes of endogenous bioactive lipids, namely glycerophospholipids/sphingolipids, classical eicosanoids, pro-resolving lipid mediators, and endocannabinoids, in AD and PD patients, as well as their most-used animal models, by means of lipidomic analyses, advocating for these lipid mediators as powerful biomarkers of pathology, diagnosis, and progression, as well as predictors of response or activity to different current therapies for these neurodegenerative diseases.

## 1. Bioactive Lipids: Main Families and Their Members

Although lipids are the major constituents of the cell membranes of all living organisms and the most efficient source of energy [1], they also act as intercellular and intracellular signaling mediators that bear many cell functions upon binding to specific G protein-coupled receptors (GPCRs) and, for this reason, are termed “bioactive lipids” [2]. Importantly, the formation and functions of these molecules strictly rely on the prevalence of omega-6 or omega-3 polyunsaturated fatty acid (PUFA) precursors and, as such, can depend on diet or be modified by supplementation. The main families of bioactive lipids are classical eicosanoids, glycerophospholipids, and sphingolipids, specialized pro-resolving mediators (SPMs), and endocannabinoids (eCBs) [3,4], as displayed in Figure 1.

Classical eicosanoids, mostly derived from the omega-6 polyunsaturated fatty acid (PUFA) arachidonic acid (AA), are the most renowned superfamily of bioactive lipids, comprising more than 120 known mediators. These include leukotrienes (LTs), prostaglandins (PGs), thromboxanes (TXs), hydroxyeicosatetraenoic acids (HETEs), epoxides (EETs), eoxins, and hepoxilins. These are involved in immune and inflammatory processes, with the aims of amplifying inflammation, coordinating leukocyte recruitment, producing cytokines and chemokines as well as presenting antigens and forming antibodies, inducing cell proliferation and migration [3]. 

Glycerophospholipids and sphingolipids comprise several compounds with great molecular diversity (e.g., phosphoinositides, phosphatidic acids, phospholipids, sphingosines, and ceramides) and with glycerol or sphingosine as respective backbones to which two fatty acids and a phosphoric acid are attached as esters [5]. The presence of an additional group attached to the phosphate allows for many different phosphoglycerides. Among the fatty acids esterified to carbon-1 and -2 (respectively denoted as the sn-1 and sn-2 positions), both saturated and unsaturated ones can occur. Typically, saturated fatty acids or monounsaturated fatty acids (MUFAs) are present at the sn-1 position, and PUFAs, such as 20:4 (AA), 22:6 (docosahexaenoic acid, DHA), and 20:5 (eicosapentaenoic acid, EPA), are present at the sn-2 position of phospholipids. This superfamily exerts pleiotropic effects, including inflammation, vesicular trafficking, endocytosis, cell cycle, cell migration and survival, as well as apoptosis and senescence [6].

The third superfamily is a relatively new class of bioactive lipids called SPMs which were identified in the laboratory of Prof. Charles Serhan. SPMs comprise over 30 different mediators that are actively synthesized during acute inflammation, either from omega-6 AA or omega-3 PUFAs, such as EPA and DHA, and they include AA-derived lipoxins (LXs); EPA-derived resolvins (RvEs); and DHA-derived resolvins (RvDs), protectins (PDs), and maresins (MaRs). They all act as immunoresolvents; that is, they stimulate the cardinal signs of the resolution of inflammation: removal, relief, restoration, regeneration, and remission [3,7,8].

The family of eCBs is a group of bioactive lipids that are able to activate type-1 and type-2 cannabinoid receptors (CB1 and CB2) and include a few lipid mediators such as N-arachidonoylethanolamine (commonly known as anandamide, AEA) and 2-arachidonoylglycerol (2-AG), as well as eCB-like 2-AG-ether (noladin ether), O-arachidonoylethanolamine (virodhamine), *N*-palmitoylethanolamine (PEA), and N-oleoylethanolamine (OEA). These molecules are produced ubiquitously by all tissues, and they serve as a homeostatic system to control several pathophysiological states and maintain human health [9].

It has been established that dysregulation of the bioactive lipid network can foster neuroinflammation and contribute to the etiopathogenesis, severity, and outcomes of many neurodegenerative and neuroinflammatory disorders. In this review, we will discuss the current literature on the quantification of the different superfamilies of bioactive lipids in the tissues and body fluids of patients affected by Alzheimer’s disease (AD) and Parkinson’s disease (PD), or their respective animal models, by the golden standard methodology of liquid chromatography or gas chromatography–mass spectrometry or mass spectrometry imaging.

## 2. Lipids and Alzheimer’s Disease

It is now evident that alterations in the brain lipid content and the composition of lipids and cerebral lipid peroxidation by genetic and environmental factors, such as apolipoprotein and lipid transporter carrying status, are key determinants for AD pathology. As a matter of fact, many studies using cell culture and transgenic animals have investigated the potential mechanisms by which ApoE4 is implicated in the pathogenesis of Alzheimer’s disease, including studies that investigated alterations in lipid metabolism, causing the inhibition of neurite extension [10]. Moreover, amyloidogenesis, an important pathogenetic factor for AD, is strictly associated with lipid composition within membrane lipid rafts, which are characterized by a combination of sphingolipids, cholesterol, saturated FAs, and a reduced content of PUFAs that serve as platforms for β-amyloid (Aβ) interactions with ApoE and tau to promote the aggregation of Aβ oligomers and hyperphosphorylation [11]. In addition, cortical and free unsaturated FAs induce the assembly of amyloid and tau filaments in vitro [12]. Over the last 30 years, all of the families of bioactive lipids have been analyzed in AD by lipidomics and will be described here and summarized in Table 1.

### 2.1. Glycerophospholipids and Sphingolipids

Abnormal phospholipid composition characterizes the brain of AD patients and mainly affects the cortex area. In the brain of AD patients, phosphatidylcholine (PC) and phosphatidylethanolamine (PE), the two most predominant phospholipids, were significantly decreased, and phospholipid diacylation products glycerophosphocholine were increased in the frontal, primary auditory, and parietal cortices [13]. Three PCs were found significantly diminished in AD: PC (16:0/20:5), PC (16:0/22:6), and PC (18:0/22:6) [14]. Mapstone et al. conducted a 5-year, observational study in healthy, elderly patients and identified ten metabolites, comprising seven PCs, one lysophosphatidylcholine, and two acylcarnitines, that were depleted in the plasma of patients with mild cognitive impairment (MCI) or AD, and the depletion could identify (with accuracy above 90%) cognitively normal individuals who, on average, will convert to MCI or AD within 2–3 years [15]. Although most studies reported a reduction of PC levels in AD, contradictory findings have also been reported. Proitsi et al. found that the lipids most strongly associated with AD were PC 40:4 and PC 36:3, both of which were increased in AD [16]. An increase in CSF PC was observed in AD brains, as compared to control brains. During normal aging, the plasma levels of lysophosphatidylcholine, choline plasmalogen, and lyso-PAF increase significantly; similar but more pronounced changes in these choline-containing phospholipids were observed in AD patients [58].

Cardiolipin is another glycerophospholipid, mainly found in the inner mitochondrial membrane, and responsible for the maintenance of the fluidity and activity of mitochondrial electron transport chain enzymes. A reduction of cardiolipin in the synaptic mitochondrial membranes was reported in the brains of AD patients [59].

Concerning sphingolipids, several studies report variations of these subtype of lipids although there is some inconsistency among them. Levels of sphingomyelins (SMs) and ceramides were studied in two different post-mortem brain regions of patients with AD. The region with extensive Aβ has increased ceramide and decreased sphingomyelin, while in the region with only diffuse Aβ deposits, the ceramide/sphingomyelin ratio was reversed [17]. In the white matter, no differences in sphingomyelin between AD and healthy subjects were found, but significant decreases in ceramide C16:0, C22:0, and C24:1 were observed [18]. Additionally, metabolomic assays reported increased SM levels in the brain tissue of AD subjects, and these were associated with the severity of AD pathology and an increased risk of abnormal cognition, including three SMs with acyl residue (SM C16:0, SM C16:1, and SM C18:1) and one hydroxysphingomyelin with acyl residue (SM (OH) C14:1) [19]. These findings came as support of former evidence in a mouse model of AD, where the intracerebral injection of Aβ promoted the catalysis of sphingomyelin to ceramide by hydrolysis and thus an increase in ceramide levels [60].

Lipidomic studies found increased ceramide levels in AD brains, in particular, ceramides Cer16, Cer18, Cer20, and Cer24 [20]. Senile plaques were abundant in saturated ceramides Cer(d18:1/18:0) and Cer(d18:1/20:0) [61]. Increased levels of ceramides were also found in the CSF [21]. Han et al. observed an enhancement of ceramide in the early AD stages in brain tissue lipid extracts by electrospray ionization mass spectrometry, while its concentration reduced with disease severity [22]. So far, few studies have measured glycerophospholipids and sphingolipids in the serum of AD patients. In particular, only three specific ceramides (containing C16:0, C20:0, and stearoyl as fatty acids) were increased, indicating that they are important predictors of cell damage and memory impairment [23]. The major metabolic product derived from ceramide is Sphingosine 1-Phosphate, and its level was decreased in the AD brain [24,62].

Over the past 15 years, it has been useful to analyze the concentration of lipids in CSF to better understand levels of brain impairment. A research team in Pasadena, California has described the presence of lipid-rich nanoparticles (NPs) in the cerebrospinal fluid of older adults [25]. During LC/MS analysis of the CSF fraction, they revealed that the NPs had a higher glycerophospholipid (PC, PE, PS, and LPS) levels than the supernatant fluid in AD patients [25]. Specifically, a subsequent study revealed that the NPs fraction composition showed increased levels of two odd-numbered MUFAs (C15:1 and C19:1) [63]. The CSF levels of sphingomyelins and phosphatidylcholine-containing MUFAs significantly correlated with p-tau levels, suggesting that this family of bioactive lipids might be associated with disease severity [64].

In search of biomarkers for the non-clinical recognition of AD stages, a method was suggested to distinguish cognitively healthy individuals with normal (CH-NAT) from pathological Aβ42/tau (CH-PAT) and AD. This study revealed that phosphatidylcholine molecular species from the supernatant fraction of CH-PAT were higher than in the CH-NAT AD participants. Furthermore, sphingomyelin levels in the supernatant fraction were lower in the CH-PAT and AD than in the CH-NAT group. The decrease in sphingomyelin corresponded with an increase in ceramide and dihydroceramide and an increase in the ceramide to sphingomyelin ratio in AD. In contrast with the supernatant fraction, sphingomyelin was higher in the nanoparticle fraction from the CH-PAT group, accompanied by lower ceramide and dihydroceramide and a decrease in the ratio of ceramide to sphingomyelin in CH-PAT, as compared with CH-NAT [26]. Other types of sphingolipids, such as sulfatides and gangliosides, although being reported to significantly vary in AD, were not the focus of the present review since they are more constituent of myelin or lipid rafts than actual bioactive lipids.

Since both glycerophospholipids and sphingolipids respectively have two or one fatty acid chains attached to the molecule of glycerol or sphingosine, a great part of literature has focused on measuring levels of FAs. The brain is highly enriched in the PUFAs DHA (22:6n-3) and AA (20:4n-6), and linoleic acid, EPA and DHA account for ~10% of brain lipids. 

Patients with MCI and AD had elevated levels of AA, but they had reduced levels of its precursor, linoleic acid (LA), as compared with healthy controls, with the latter progressively decreasing in cognitive performance [27]. Oleic acid (OA), an omega-9 FA and the most abundant dietary FA, was decreased in the frontal cortex and hippocampus of AD brains [28]. Furthermore, a case-control study performed on 148 AD patients reported significantly lower serum levels of DHA, and this reduction was associated with the severity of clinical dementia [65]. A more complete study analyzed the level of FFAs in the serum of AD patients and found that several of them significantly decreased when compared to the control; the study included 3 saturated fatty acids (C14:0, C16:0, and C18:0) and 6 unsaturated fatty acids (C16:1, C18:1, C18:2, γ-C18:3, C20:2, and C22:6). The serum level of C18:3 was significantly higher in AD patients [30]. In an elegant study by Martin et al., lipid composition was analyzed in lipid rafts of human frontal brain cortex obtained between 3- and 18-h post-mortem, revealing that lipid rafts from AD brains exhibit aberrant lipid profiles compared to healthy brains. In particular, lower levels of omega-3 long-chain PUFAs (mainly DHA) and oleic acid were observed [31].

Studies testing six unsaturated FAs, including linoleic acid (LA), AA, α-linolenic acid (ALA), DHA, EPA, and OA, showed that all these unsaturated FAs were positively associated with neuritic plaques and neurofibrillary tangle burdens, and they were negatively correlated with cognitive performance. In brain regions vulnerable to AD pathology—the middle frontal and inferior temporal gyri, there were decrements in LA, ALA, and AA, and increases in DHA [32]. All these unsaturated FAs can directly interact with Aβ40 and Aβ42 peptides, and they display excellent anti-aggregation properties by preventing amyloid fibril formation, especially OA and DHA [66]. However, when investigating the role of different unsaturated FAs in modulation of neuroprotective α-secretase-cleaved soluble APP (sAPPα) secretion and cell membrane fluidity, only AA, EPA, and DHA with four or more double bonds are capable of increasing membranous fluidity and sAPPα secretion, whereas stearic acid (SA, 18:0), LA, ALA, and OA cannot [67].

Among all PUFAs, DHA is assuredly the most investigated in AD, and since the end of the 1990s, many studies have concurred that a reduction of this lipid in several tissues [10,28] suggests a key role in AD pathology. AD patients have decreased DHA levels throughout their brains, including the disease-resistant regions, but the most prominent reduction occurs in the hippocampus, and DHA content has been found to show a positive correlation with dementia [33]. Livers from AD patients also contain lower levels of DHA but higher levels of short-chain n-3 precursors, including tetracosahexaenoic acid, suggesting a defect in its bioconversion into DHA [34]. However, some studies reported no significant changes in DHA levels between erythrocytes or the brain tissue of AD and control subjects [68,69].

In contrast, a very recent study with 2 years of follow-up reported that DHA is a strong protective factor for cognitive decline, inasmuch as AD patients who had stable cognitive impairment for 2 years resulted in higher baseline serum DHA levels than patients with declining AD [70].

### 2.2. Classical Eicosanoids

Most of the eicosanoids derived from arachidonic acid, i.e., PGs, LTs, HETEs and EETs, have been detected and associated with AD, as elegantly reviewed by Biringer [40] and reported by other studies that analyzed their oxidation products [71] or revealed specific increases in 12-HETE in plasma and brain and decreases in 15-HETE and PGD_2_ during the early phase [35]. An eicosanoid known to be a specific marker of in vivo lipid peroxidation is isoprostane-F2a, which is present in elevated concentrations in the CSF of AD [36] and in plasma, too [37,72]. Although plasma levels of F2a-isoprostanes do not qualify as robust biomarkers for AD diagnosis as in the CSF, even if plasma measurements may still have value in clinical trials. A hypothetical classification system for AD diagnosis is based on CSF, Aβ42, and tau levels, which allowed the improvement of the diagnostic accuracy for AD. For discriminating AD from non-AD, the level of CSF F2-IsoP needs to be greater than 25 pg/mL [38]. However, in another study, it was found that peripheral F2A and F4-Neuroprostatene levels in urine and plasma are not increased in AD [73,74]. 

The alteration of neurotransmitters, lipids, and sterol metabolism are the pathways impaired in the CSF of AD patients. The major metabolites altered are prostaglandins (PGG_2_ and PGJ_2_), hydrocortisone, and tetrahydrocortisone [39]. In a recent work on the CSF of AD patients, the levels of both pro-inflammatory and pro-resolving LMs seem to be associated with cognitive dysfunction [50]. 

Although analyses of leukotrienes in CSF have been performed since the 1980s, no significant alteration in the context of AD is known. The levels of PGs in post-mortem brains revealed regional differences in the patterns of PG profiles between the parieto-occipital cortex and the other cortical areas; only TBX_2_ and PGD_2_ were greater than in controls [41]; specifically, PGD_2_ was significantly increased in the frontal cortex [40], and PGE_2_ and PGF_2_ were reduced [29].

Regarding those fluid samples more easily accessible from the patient, such as CSF, plasma, and urine, the levels of eicosanoids were significantly different from those of control subjects. Many studies observed alterations in the levels of diacylglycerols, prostaglandins, and phospholipids, which can be related to oxidative stress and membrane breakdown.

Elevations in serum prostaglandin levels are important markers of oxidative stress [75]. PGE_2_ concentrations in CSF were approximately 5 times higher, and 6-keto-PGF1a levels were 4-fold less than those of control subjects [42]. In contrast, the detection of PGF_2a_, PGD_2_, and TXB_2_ did not reveal differences between AD and the control. Subsequently, a study investigated the correlation between PGE2 and cognitive scores, and they found that PGE2 levels were higher when learning scores were just below the normal range (early Alzheimer’s disease), but declined with progressive learning impairment [39,43,76]. These findings support the hypothesis that inflammatory processes predominate in early AD.

In urinary samples, the concentrations of F2-isoprostanes, PGF_2_ and 8-isoPGF_2_, significantly increased in AD patients when compared to those of the healthy subjects [44]. Thromboxane (TX), the last class of eicosanoids, did not seem to change in plasma or CS; instead, urinary samples with higher concentrations of 11-dehydro-TXB_2_ were reported in AD patients, as compared to healthy controls [77]. Concerning this lipid class, an Alzheimer’s Disease Anti-inflammatory Prevention Trial (ADAPT) analyzed the urine and plasma of participants to determine whether treatments for disease produced serious, adverse cardiovascular (CV) events, and an association between high urine Tx-M/PGI-M ratios and CV events was observed [78].

### 2.3. Specialized Pro-Resolving Mediators

Despite the relatively recent discovery of this superfamily of bioactive lipids, which is constantly flourishing and up-to-date counts include more than 30 members, measurement and quantification of SPMs in AD started as early as 2005 with the work by the group of Prof. Nicolas Bazan, when the levels in post-mortem AD brain tissues of the DHA-derived neuroprotectin D1 (NPD1) were found to be reduced in the post-mortem hippocampal CA1 region, but not in the thalamus or occipital lobes of AD patients, as compared to age-matched controls [45]. After this study, the reduction of NPD1 levels during the AD course have been corroborated in the hippocampus of the 3xTg mouse model of AD [46]. Although only two studies used the less-standardized enzyme immune assay (EIA) developed for a few SPMs by Cayman, whereby LXA_4_, RvD2, and RvE1 were reduced in the brain or hippocampus of 3xTg mice and 5xFAD mice, respectively [79,80], the first complete lipidomic profiling was undertaken in 2015 by the group of Prof. Serhan. This study found a reduction of RvD1 in CSF but not in the hippocampus, and it found decreased levels of MaR1 in the hippocampus in post-mortem human AD brains [47]. The same group conducted randomized, double-blind, and placebo-controlled clinical trial on AD patients—the OmegAD study—in which a placebo or a supplement of 1.7 g DHA and 0.6 g EPA was taken daily for 6 months. After the treatment, RvD1 and LXA_4_ levels were measured by EIA on the peripheral blood mononuclear cells (PBMCs), and their levels remained unchanged, suggesting that omega-3 supplementation prevented a reduction in SPMs [81]. A similar result was obtained in another clinical trial in human patients affected by mild cognitive impairment (MCI) and pre-MCI conditions, whereby the supplementation with omega-3 fatty acids and antioxidants led to increased levels of RvD1 in macrophage cultures isolated from PBMCs, as quantified by EIA [82]. The search for SPM measurements as potential biomarkers of AD pathology was then performed on a brain area different from the previously analyzed hippocampus, namely the entorhinal cortex, an anatomical region of importance for memory which is affected early in AD pathogenesis. In this study, several SPMs were significantly reduced as compared to healthy controls, such as PD1, MaR1, and RvD5, while LXA_4_, LXB_4_, RvD1, RvD2, RvE1, and RvE2 did not show relevant changes [48]. Subsequently, SPMs were analyzed in 3 different mouse models of AD: Fat-1 mice, Tg2576 mice, and APP/PS1/SphK1 mice. As expected, these studies led to contrasting results, and several SPMs were undetectable in the brain [83], increased in the plasma [84]—in both cases following a DHA-enriched diet, or reduced in neuronal cells [49]. 

Recently, the first lipidomic profiling of cerebrospinal fluid in AD was undertaken to simultaneously analyze both SPMs and classical eicosanoids in patients with cognitive impairment, ranging from subjective impairment to a diagnosis of AD, and correlated to cognition, CSF tau, and β-amyloid. In this study, RvD4, RvD1, PD1, MaR1, and RvE4 were lower in AD and/or MCI, as compared to patients with spinal cord injuries. Furthermore, levels of RvD1 showed a negative correlation with p-tau levels, while RvD4 negatively correlated with AD tangle biomarkers, and positive correlations with cognitive test scores were observed for both SPMs and their precursor fatty acids [50].

### 2.4. Endocannabinoids

Despite eCBs having been discovered in the early 1990s, very few reports have measured their levels in AD patients. The first report was in 2006, but that was in a rat model of AD injected with Aβ that reported an increase in the hippocampal levels of 2-AG, but not AEA [51]. This was confirmed 10 years later in the more-commonly used APP/PS1 transgenic mouse model of AD, in which 2-AG levels were increased in the cortex although that occurred upon inhibition of its degrading enzyme, monoacylglycerol lipase (MAGL) [52]. The first LC-MS/MS study on eCBs levels on AD patients was not performed until 2009. Here, no differences in the plasma levels of 2-AG and AEA were found between AD patients and healthy controls. However, only 2-AG—but not AEA—was detected in the CSF of these patients, and its levels, although they did not correlate with cognitive performance in healthy controls at risk for AD, showed an inverse correlation with TNF-alpha [85]. Moreover, 2-AG levels were found to be increased in blood samples of AD patients, which nonetheless showed no differences in other eCBs, such as AEA, PEA, or OEA [53].

LC-MS/MS analyses of the brain areas of post-mortem AD patients revealed no variations in 2-AG or PEA, but there was a significant reduction in AEA and its precursor 1-stearoyl,2-docosahexaenoyl-sn-glycerophosphoethanolamine-N-arachidonoyl (NArPE) in midfrontal and temporal cortex tissue, but not in the cerebellum [54]. Over the following 5 years, levels of eCBs were only investigated in murine models of AD. A metabolomic brain profiling of PS1/APP AD mice was performed and revealed an increase in both AEA and 2-AG, as compared to their wild-type littermates, especially for 2-AG in mice that also carried a genetic deficiency of MAGL [55]. A similar result was obtained in another genetic model of AD, where 2-AG increased in the brain of 5XFAD upon administration of a potent MAGL inhibitor [56]. Another study on the AβPPswe/PS1ΔE9 mouse model quantified eCB levels and assessed lipidomic profiles in the frontal cortex, hippocampus, and striatum tissues to determine whether regional variations would be observed with age and disease progression. This study showed age-related increased levels of AEA, PEA, and OEA in the hippocampal and frontal cortex tissues of both AD and control mice, but the hippocampi of the former had higher concentrations of AEA than in those of the latter, while in the striatum, lower levels of 2-AG have been reported [57]. More recently, a new study quantified the AEA and 2-AG in the human plasma of AD patients by means of a novel, selective column-switching ultra-high performance liquid chromatography–tandem mass spectrometry (UHPLC–MS/MS) method, allowing for a faster approach with fewer steps [86]. Although the values for both AEA and 2-AG agreed with data previously reported in the literature, their concentration ranges were smaller.

## 3. Lipids and Parkinson’s Disease

One of the neuropathophysiological hallmarks of PD is characterized by the accumulation of α-synuclein (α-syn) and the formation of filamentous aggregates called “Lewy bodies” in the brainstem, limbic system, and cortical areas, leading to a progressive loss of dopaminergic neurons in the substantia nigra and striatum, and thus motor dysfunction [87]. Although the etiology of PD is generally unknown, the formation of α-syn aggregates seems to be closely associated to an altered lipid metabolism. Indeed, despite the fact that the physiological role of α-syn is yet not fully understood, it appears that its accumulation and aggregation are either enhanced by membrane lipid composition or they further cause alterations in lipid homeostasis [88]. Furthermore, genome-wide screening studies in yeast support the involvement of lipid metabolism in α-syn toxicity, with many genes associated with lipid metabolism, modifying toxicity, and vesicle-mediated transport [89]. Additionally, many known PD-risk genes, such as glucocerebrosidase (GBA), leucine-rich repeat kinase 2 (LRRK2), and parkin (PARK2), show a clear association with lipid metabolism, and particularly lipid accumulation [90,91,92]. Modern lipidomic approaches further underline the significant roles of lipids, and particularly dysfunctions in lipid metabolism, in the pathogenesis of PD, which are discussed in this review and summarized in Table 2.

### 3.1. Glycerophospholipids and Sphingolipids

Similar to AD, PD changes in the lipid membrane composition are strictly associated with its pathology, and superfamilies of several bioactive lipids seem to be involved in the pathogenesis of PD [133,134].

One prominent gene involved in the pathogenesis of PD is the GBA gene, which encodes for the β-glucocerebrosidase enzyme, which hydrolyzes glucosylceramide. Mutations in this gene are found in up to 10% of patients with PD, thus implicating aberrant sphingolipid metabolism [135]. Indeed, the CSF lipidome signatures of Parkinson’s patients at early stages, de novo PD patients with abnormal dopamine transporters, and healthy controls show distinct lipidome changes, with a significant increase in glucosylceramide (GlcCer), while sphingomyelin (SM) was significantly reduced in GBA-PD patients [93]. Furthermore, applying the ratio of GlcCer to SM for stratifying cases of idiopathic PD revealed that patients with a high GlcCer/SM ratio present increased cognitive deterioration compared to those in the lowest quartiles. However, other studies did not find any significant accumulation of GluCer in the CSF of PD subjects [136]. Of note, PD patients with neuropathic and dyskinetic pain had higher plasma levels of GlcCer compared to PD patients with no sensory pain [116]. 

Interestingly, plasma ceramide lipidomics linked cognitive impairments among PD patients with higher levels of the plasma ceramides C16:0, C18:0, C20:0, C22:0, and C24:1, and the monohexosylceramide species C16:0, C20:0, and C24:0 [94]. Thus, plasma ceramide and monohexosylceramide metabolism seem to be altered in PD and also in non-GBA mutation carriers, with higher levels being accompanied by worse cognition. These findings were confirmed by shotgun lipidomics of L444PGBA mutation fibroblasts, an ex vivo PD system that presents a mutation of the GBA gene, showing distinct, increased proportions of ceramides and hexosylceramide, while total phospholipids were significantly decreased [95]. Intriguingly, isolated membrane lipids from PD-GBA cultures proved to be more potent triggers of recombinant human α-syn in vitro fibrillation compared to lipids from healthy controls [95]. LRRK2 is another important gene for the development of PD, and it has been implicated in a variety of pathways, including mitochondrial dynamics. LRRK2 is mutated in families with autosomal-dominantly inherited PD, and it has been acknowledged as a susceptibility factor for PD [137]. Although in vivo models of LRRK^−/−^ show an intact dopaminergic system [138], targeted lipidomics approaches identified altered sphingolipid composition in LRRK2^−/−^ mouse brains, with significantly increased levels of Cer in LRRK2^−/−^ mice and direct alterations in GBA1 enzymatic activity [96].

Furthermore, new machine learning analysis has been introduced to link lipidome data of PD patients to the course of disease severity by using whole blood lipidomics and an established lipid panel containing dihydrosphingomyelin (dhSM), GlcCer, dihydro globotriaosylceramide (dhGB3), and dihydro GM3 ganglioside (dhGM3). [139]. In addition, untargeted lipidome assessments were utilized to detect incipient dementia in PD. Lipidome analysis by isotope-standard-assisted liquid chromatography of PD patients’ serum revealed that an association with patients transitioning to the development of dementia could be characterized by a 5-lipid biomarker panel [97]. 

A multi-omic integration analysis on 30 drug-naive, de novo PD patients and 30 matched controls revealed that a longitudinal trajectory with 3 long-chain fatty acids (FA) (FA C14:0, FA C17:1, and FA C20:1) and dopaminergic medication exerted a strong change in lipidomic signature, with a prevalence of phospholipid species and breakdown products of phospholipids [98]. 

The substantia nigra forms the major spot of interest in PD research, as this brain area is characterized by the loss of dopaminergic neurons, resulting in the clinical manifestation of PD. However, information regarding the lipid alterations in established PD animal models is still scarce. HPLC-ESI-MS/MS measurements of the substantia nigra from rats 21 days after an infusion of 20 µg of 6-OHDA, or a saline vehicle into the anterior dorsal striatum, displayed a relative down-regulated abundance of several PC species, while known neuroinflammatory signaling lipids, namely lysophosphotidylcholine (16:0 and 18:1), were up-regulated [99]. 

In post-mortem human substantia nigra samples, 5 lipid species have shown different abundances between PD patients and controls, namely Bis (Monoacylglycerol) Phosphate (BMP) 42:8, PC 36:3, PE A36:2, PI 42:10, and PS 36:3, with BMP and PI showing a significant upregulation in PD samples [100]. The same study also showed a saturated sphingomyelin species depletion in the putamen of PD patients. Post-mortem-acquired sphingolipidome data of the caudate, putamen, and globus pallidus revealed altered levels of sphingolipid species, including ceramides (Cer), dihydroceramides (DHC), hydoxyceramides (OH-Cer), phytoceramides (Phyto-Cer), and phosphoethanolamine ceramides (PE-Cer) in the PD subjects compared to healthy controls. The putamen showed the strongest effect of depletion of Cer and Cer-OH, while sphingomyelin levels remained largely unchanged across the basal ganglia of both PD and control samples [101]. These results are in line with prior studies reporting unchanged total glucosylceramide levels in the putamen and cerebellum in both GBA-PD and sporadic PD brain samples [140]. However, other studies have reported decreased levels of SM in the anterior cingulate cortex [102] and increased levels in the substantia nigra [103], while lipidome analysis of the visual cortex, a brain area that does not show any formation of Lewy bodies in PD, showed increased overall sphingolipid levels [104]. Applied lipidomics on the Lewy bodies in the post-mortem mesencephalon and *corpus callosum* of PD patients with dementia confirmed a high cell-membrane-related lipid content within the Lewy bodies, with the mass spectra showing strong peaks corresponding to SM and PC for positive immunostained α-syn inclusions isolated from both the SN and hippocampal CA2 sector [141]. However, the profile of SM/PC does not seem to be specific for LB. 

It is important to note that, while post-mortem data acquired from PD human brain samples show distinct changes in lipid profiles, post-mortem-acquired parkinsonian CSF seems to present increased lipid levels of almost the complete profile of PC and the majority of CM and SM [105]. This effect might result from the breakdown of the blood–brain–barrier and the subsequent influx of non-CSF lipids into the CSF, thus potentially leading to an impaired lipid homeostasis. 

Although most lipidomics approaches in PD are focused on the CSF and blood, lipidomics have also been introduced for the examination of other biological substances and tissues, such as the sebum or skin fibroblasts. Sebum is an oil-like substance produced by the sebaceous glands and is predominantly composed of triglycerides, fatty acids, wax esters, squalene, and cholesterol. LC-MS analyzation of sebum samples, including drug-naïve PD, medicated PD, and healthy controls, revealed an enrichment of the sphingolipid metabolism in both drug-naïve and medicated PD [106]. Lipid dysregulation in the sphingolipid metabolism in the sebum of PD patients reflected an overall lipid dysregulation. Additionally, lipid profiling of parkin-mutant, human skin fibroblasts also found these lysophophatidylcholine species to be upregulated [107]. Also, while a majority of the presented lipidomic data were acquired by mass spectrometry, different lipidomic approaches involving lipoprotein profiling by the implementation of NMR spectroscopy, bundled with multivariate data pre-processing, show promising results in discriminating between early-stage PD and PD-related dementia [142].

### 3.2. Classical Eicosanoids

Higher PUFA levels were detected in the supernatants and high-speed membrane fractions of neuronal cells over-expressing wild-type or PD-causing mutant α-syn. This increased PUFA content in the membrane fraction was accompanied by increased membrane fluidity in the α-syn overexpressing neurons. Membrane fluidity and the levels of certain PUFAs were lower in the brains of mice that have had α-syn genetically deleted [108]. Furthermore, lipid rafts were purified from human frontal cortex from the normal, early motor stages of PD and incidental PD subjects, and the lipid composition was analyzed. Lipid classes were separated from a fraction of total lipid by one-dimensional, double-development, high-performance, thin-layer chromatography. The results show that lipid rafts from both cohorts of patients exhibited dramatic reductions in their contents of omega-3 and omega-6 long-chain PUFAs, especially DHA and AA [109]. 

Metabolomic approaches using UPLS-MS measured the plasma levels of 158 fatty-acid metabolites in a cohort including 42 PD patients and 54 healthy volunteers. Results showed an increase in 2 eicosanoids (arachidonic acid and 13-hydroxy-octadecatrienoic acid) and a decrease in 11 eicosanoids (DHA, lyso-platelet-activating factor, 12-hydroxy-eicosatetraenoic acid, dihydroxy-eicosatrienoic acids, dihidroxy-octadecenoic acids, 17,18-dihydroxy-eicosatetraenoic acid, and hydroperoxy-octadecadienoic acids) in PD patients compared to healthy controls [110]. A similar study identified LTB_3_ as a potential biomarker for PD [111].

Among PGs, PGE_2_ is the most associated with PD, and several lipidomic studies identified this eicosanoid in PD patients. Indeed, PGE_2_ levels were increased in the substantia nigra [143] and the CSF of PD patients with mild cognitive impairment and dementia although, in the first study, it was measured by enzyme immunoassay, whereas in the second study, its levels were not associated with elevated total-tau, phosphorylated-tau, or other inflammatory mediators [144]. In an α-syn model of PD, among all prostaglandins assayed in the brain (PGE_2_, PGD_2_, and PGF_2α_), PGD_2_ showed the greatest increase (two-fold) compared to wild-type animals [112]. Furthermore, analysis of brain extracts by electrospray ionization–Fourier transform resonance mass spectrometry (ESI–FT-ICR-MS) revealed an increase in two different prostaglandins (PGB_1_ and PGH_2_) and in 15(*S*)-HETE in brain extracts of a Parkinson-like, rat model chronically exposed to manganese [113]. 

Other eicosanoid members, such as isoprostanes and HETEs, were also identified in PD. Indeed, the levels of plasma F_2_-isoprostanes (F_2_-IsoPs), neuroprostanes (F_4_-NPs), and HETEs were found to be higher in PD patients compared to controls, especially in the early stages of PD. In the same study, a significant negative correlation between the cumulative intake of L-DOPA and plasma total HETEs was also observed [114]. Changes in PGE_2_ and PGD_2_ levels were corroborated in other models of PD. Indeed, while mice carrying the mutated disease gene DJ-1 showed reduced PGE_2_ and PGD_2_ levels [145,146], LPS-injected mice displayed increased PGE_2_ levels [147,148], and MPTP-injected mice presented enhanced striatal and nigral levels of LTB_4_ [149]. However, 6-OHDA-treated rats showed lower PGE_2_ levels 4 weeks after treatment compared to controls [150]. Of note, in these studies, eicosanoids were always measured with the commercially available enzyme immunoassay kits from Cayman, and not by lipidomics. 

Additionally, LC/MS analysis in rotenone-treated rats showed a significant reduction of oxidizable PUFA-containing phospholipid cardiolipin species and increased contents of mono-oxygenated CL species in later stages of exposure in the substantia nigra. Notably, linoleic acid in the sn-1 position was the major oxidation substrate yielding its mono-hydroxy- and epoxy-derivatives, whereas more readily “oxidizable” fatty-acid residues, AA and DHA, remained non-oxidized in the substantia nigra. On the other hand, elevated levels of PUFA CLs were detected in plasma [115].

### 3.3. Specialized Pro-Resolving Mediators

As for this superfamily of bioactive lipids, so far, only 1 study from our group measured 2 specific SPMs, i.e., RvD1 and RvD2, in both the plasma and CSF of the transgenic and over-expressing α-syn (Syn) rat model of PD, as well as in early-PD patients. In particular, although no changes were observed in RvD2 levels, the content of RvD1 was higher in 2-month-old Syn rats and progressively decreased in 4-month-old and 18-month-old rats compared to their wild-type littermates, suggesting an impairment in the production of this SPM along disease progression, starting at an early phase when neuroinflammatory, synaptic, and motor-behavioral dysfunctions started to appear. Early-onset PD patients also showed strong reductions in RvD1 levels in both plasma and CSF [128]. However, the measurement of RvD1 and RvD2 were performed by commercially available EIA kits from Cayman, and so far, no study has ever attempted to measure all of the other SPMs through targeted-lipidomics analysis.

### 3.4. Endocannabinoids

Studies on the quantification of eCBs in PD are numerous. The very first report of eCB analysis in PD came from the group of Prof. di Marzo, in which AEA and 2-AG were measured in two regions of the basal ganglia in the reserpine-treated rat model of PD, namely the globus pallidus and the substantia nigra. Although the levels of AEA in these anatomical regions were three-fold higher than those previously reported in any other brain region, only 2-AG showed a strong increase in the globus pallidus compared to control animals [117]. In another rat model of PD, induced by a unilateral nigral lesion with 6-hydroxydopamine (6-OHDA), the levels of AEA were increased in the striatum 6-OHDA-lesioned rats compared to control animals, whereas endogenous 2-AG was unaffected. The authors suggested that this was attributable to decreased AEA degradation, rather than an increased synthesis [118]. This was supported by similar findings observed by the group of Prof. Maccarrone, in which, once again, only AEA striatal levels were increased in 6-OHDA, and these were reversed upon chronic L-DOPA treatment [119]. In the same year, another study investigated the effect of L-DOPA on eCB levels in 6-OHDA rats and found that systemic administration of L-DOPA increased anandamide concentrations in the basal ganglia via activation of dopamine D1/D2 receptors, whereas its levels were significantly reduced in the caudate–putamen ipsilateral to the lesion. These results indicate that a deficiency in eCB transmission may contribute to levodopa-induced dyskinesias [120]. However, a similar study reported years later that both striatal AEA and 2-AG were reduced in these 6-OHDA rats upon L-DOPA administration [121], suggesting that further studies on other LID models and human samples are still required to validate the mechanism of endocannabinoid in LID.

A more detailed L-DOPA treatment was undertaken in the MPTP-lesioned, non-human primate model of PD, where cynomolgus monkeys were either acutely treated (non-dyskinetic) or long-term treated with L-DOPA (levodopa-induced dyskinesia, LID). In the untreated, MPTP-lesioned primates, parkinsonism was associated with increases in both 2-AG and AEA in the striatum, and in only 2-AG in the substantia nigra. Increased levels of AEA in the external globus pallidus of MPTP-lesioned animals were normalized by L-DOPA treatment and may contribute to the generation of parkinsonian symptoms. However, no evident alterations in eCB levels were associated with the expression of LID [122]. Levels of 2-AG were also found to be increased in MPTP-lesioned mice [123]. Following MPTP administration, levels of 2-AG in the ventral midbrain started to increase 2 days, and up to 4 days, after MPTP injection and returned close to control levels after 7 days, whereas in the striatum, 2-AG levels were reduced after 7 days [124]. Thus, changes in 2-AG were time- and region-specific following MPTP administration, indicating that eCBs represent a natural defense mechanism against inflammation. Subsequently in neurotoxic and inflammation-driven rat models of PD with striatal injections of 4 different toxins/inflammatory stimuli (6-OHDA, LPS, rotenone, and Poly I:C), not only AEA and 2-AG, but also other eCB-like members, such as PEA and OEA, were measured for the first time. Interestingly, while LPS caused significant increases in all 4 eCBs at specific time-points, 6-OHDA was only able to induce PEA and OEA, but not AEA or 2-AG, in the striatum [125]. On the other hand, rotenone injection led to an increase in the striatal levels of AEA, PEA, and OEA, while viral stimulus Poly I:C induced PEA and OEA, but not AEA or 2-AG, albeit at specific time-points [126]. These findings suggest that changes in eCB levels may be functionally important in inflammation-driven neurodegeneration, rather than in direct neurotoxicity per se. Despite the fact that changes in eCB levels were indeed observed in several neurotoxic models of PD, no variation in AEA or 2-AG was observed in the striatum of PINK-deficient mice, a gene involved in early-onset PD [129]. However, in yet another PD animal model overexpressing α-syn, eCBs not only were significantly reduced (especially 2-AG and OEA) but such reduction was manifested not at an early stage, but at an advanced stage, suggesting that α-syn overexpression triggers a dysregulation of eCB biosynthesis [127].

The very first report on eCB measurement in PD patients was performed in 2005 by the group of Maccarrone in the CSF of 16 patients at different stages of disease, from de novo PD diagnosis to early-mild or advanced, and all had undergone a complete drug washout before CSF examination. AEA levels showed a 2-fold increase in PD patients compared to control subjects, and this increase was independent of the stage of disease [151]. The same group extended the study in PD patients to include subjects undergoing drug withdrawal and those under pharmacological therapy. In untreated patients AEA levels were more than doubled as compared to controls, and the levels were restored following chronic dopaminergic replacement [152,153]. Based on these studies, the authors suggested that the increase in AEA might reflect a compensatory mechanism occurring in the striatum of PD patients, aimed at normalizing chronic dopamine depletion. Plasma and CSF levels of eCBs were also measured in PD patients, with and without LID, and LC-MS/MS analysis showed elevated CSF levels of AEA and decreased CSF and plasma levels of 2-AG, as compared to healthy controls, and these changes were curiously observed only in patients without LID [130]. Of note, AEA levels were reduced in PD patients with ongoing pain and showed a linear relationship with pain intensity and sensory losses [116].

More recently, several groups focused on developing novel mass-spectrometry approaches to measure eCB in PD patients or animal tissues. For instance, a study described the development and validation of an ultra-high performance liquid chromatography–tandem mass spectrometry method that uses disposable pipette extraction (DPX-UHPLC–MS/MS), and this novel approach was successfully applied to determine AEA in CSF samples of PD patients, providing a simpler, faster, and highly sensitive procedure that requires smaller CSF sample and organic solvent volumes [131]. The same group developed 2 additional simple and reliable methods to determine AEA and 2-AG levels in the brain hemispheres of 6-OHDA rats: (i) by micro salting-out assisted liquid–liquid extraction combined with ultra-high performance liquid chromatography–tandem mass spectrometry (SALLE/UHPLC-MS/MS) [154] with no significant matrix effect, and inter- and intra-assay precision and accuracy with CV and RSE values lower than 15%, respectively; and (ii) by in-tube solid-phase microextraction (in-tube SPME) directly coupled to a tandem mass spectrometry (MS/MS). This showed similar advantages to the first approach, but the selectivity of the in-tube SPME and MS/MS (MRM mode) techniques allowed them to be directly coupled online, which dismissed the need for the chromatographic separation step [132]. Due to space limitations, studies measuring eCB levels in different models of PD upon the pharmacological or genetic modulation of their respective degrading enzymes are not discussed here.

## 4. Concluding Remarks

Changes in the lipidomic profiles of the different superfamilies of bioactive lipids are apparent in biological fluids, such as plasma and CSF, as well in specific anatomical brain regions that are associated with the pathological hallmarks of both AD and PD. With the exception of SPMs in Parkinson disease, where no targeted-based lipidomic has ever been performed, it is now clear that dysregulated lipid homeostasis in several members of glycerophospholipids/sphingolipids, classical eicosanoids, SPMs and endocannabinoids (only for AD) contributes greatly to the pathogenesis and even progression of these neurodegenerative diseases. Since each superfamily of these bioactive lipids has a distinct role in either sustaining the neuroinflammatory and degenerative processes or in reducing them, it is important that future studies apply integrated multi-omic approaches that will be able to simultaneously measure all of them in the same sample in order to have a clear metabolic fingerprint and profile of the inflammatory status of each patient. This is undermined by the limited availability of human samples, and novel mass-spectrometry implementations, which can reduce the amount of a sample needed to few microliters to detect all lipid mediators, will be crucial. 

In most cases, the detection of certain lipid mediators in plasma or CSF could be potentially useful for monitoring the diagnosis, progression, and prognosis of the disease. However, confounding factors form a major issue in lipidomics analysis with gender-, gut-microbiota-, and diet-induced alterations in lipid composition affecting the identification of disease- and therapy-specific lipidome signatures. Most importantly, modern mass spectrometry technologies provide quantitative readouts, but many studies do not report absolute lipid concentrations and differ vastly in methodology, even in regard to lipid extraction, workflow, and data presentation. Hence, a network of lipidomic standards is highly encouraged in order to minimize interlaboratory differences and to prove its clinical use [155].

## Figures and Tables

**Figure 1 ijms-23-06235-f001:**
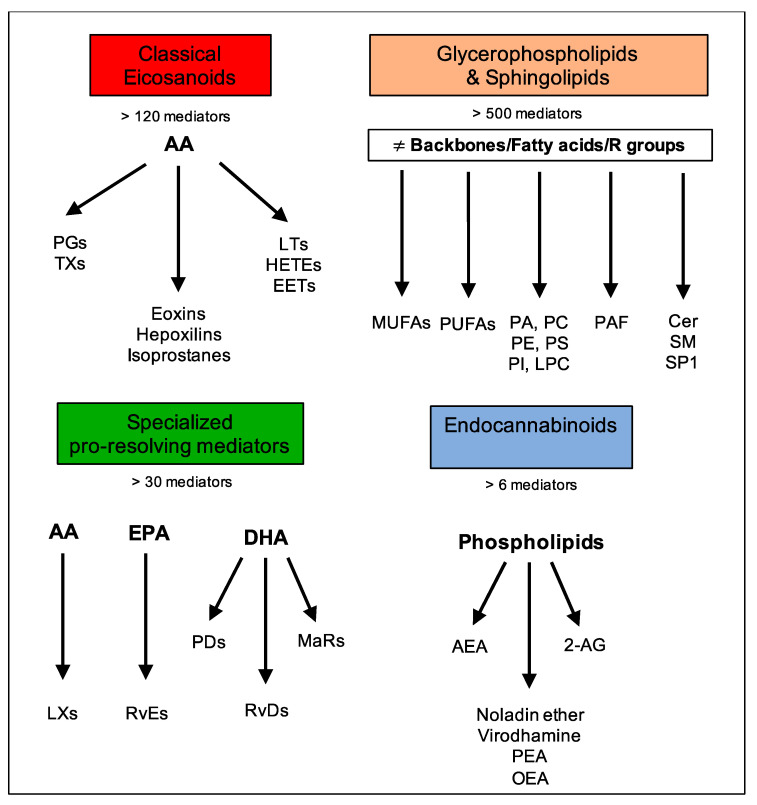
Main families of bioactive lipids and their mediators.

**Table 1 ijms-23-06235-t001:** Bioactive lipid evaluation in Alzheimer disease by lipidomics.

Bioactive Lipid	Methods of Detection/Instruments	Tissue	Variation	References
PCs and PEs	HPLC Microsorb C-18 column (Rainin Instruments) and detected fluorimetrically	Human frontal cortex	Decreased levels in AD patients	[13]
PC (16:0/20:5), PC (16:0/22:6), and PC (18:0/22:6)	LC/MS ACQUITY UPLC™ system (Waters Inc.) and Bruker Advance 600 spectrometer	Human plasma	Decreased levels in AD patients	[14]
PC diacyl (C36:6), C38:0, C38:6, C40:1, C40:2, C40:6, PC acyl-alkyl C40:6, lyso-PC (C18:2), acylcarnitines and C16:1-OH	LCMS/MS (Biocrates Absolute-IDQ P180)	Human plasma	Decreased levels in MCI and AD patients and positive correlation with MCI/AD conversion	[15]
PC 40:4 and PC 36:3	LC/MS ACQUITY UPLC and XEVO QTOF system (Waters Inc.)	Human plasma	Decreased levels in AD patients	[16]
SMs and ceramides	ES/MS/MS Sciex API 3000 triple stage quadrupole tandem mass spectrometer	Human brain	SMs decreased and ceramide increased in brain regions of AD patients with extensive Aβ SMs increased and ceramide decreased in brain regions of AD patients with diffuse Aβ	[17]
Ceramide (C16:0, C22:0, and C24:1)	LC-MS Ultra-high performance liquid chromatography C8-reversed phase column (BEH C8, Waters) connected to an Orbitrap Exactive mass spectrometer and analyzed once in positive and once in negative ion mode	Human white matter	Decreased levels in AD patients	[18]
SM (C16:0, C16:1, C18:1, and C14:1)	HPLC-MS/MS AB SCIEX 4000 QTrap mass spectrometer	Human blood, brain, and CSF	Increased levels in AD patients and positive correlation with disease severity	[19]
Cer16, Cer18, Cer20, and Cer24	HPLC-ESI-MS/MS Triple quadrupole 6410 mass spectrometer (Agilent Technologies) and ionization in the positive ion mode	Human brain and CSF	Increased levels in AD patients compared to other neurological diseases	[20,21,22,23]
S1P	HPLC HPLC system Monitored using a model 474 scanning fluorescence detector	Human brain	Reduced levels in AD patients	[24]
PC, PE, PS, and MUFAs (C15:1 and C19:1)	LC-MS LTQ ion trap mass spectrometer equipped with a nano-electrospray ion source (Thermo Fisher Scientific) and 30 μm PicoTip emitter (New Objective)	Human NPs fraction of CSF	Increased levels in AD	[25]
PC	ESI-MS TSQ mass spectrometer (Thermo Fisher Scientific)	Human CSF	Decreased levels in AD patients with normal Aβ42/tau compared to pathological Aβ42/tau	[26]
AA and LA	GC-MS Agilent 7820A Plus Gas Chromatograph	Human plasma	Increased levels in MCI and AD patients compared to controls, by contrast no differences are noticed between AD and MCI	[27]
OA, PGE_2_, and PGF_2a_	TLC-HP-FFAP and TLC Hewlett Packard gas chromatograph using a capillary column	Human frontal cortex and hippocampus	Increased levels in AD patients compared to other cortical areas	[28,29]
SFAs (C14:0, C16:0, and C18:0) and UFAs (C16:1, C18:1, C18:2, γ-C18:3, C20:2, and C22:6)	GC-MS Gas chromatograph coupled to an ion trap mass spectrometer (Thermo Finnigan)	Human serum	Decreased FFAs levels in AD patients Increased C18:3 in AD patients	[30]
DHA and OA	HPTLC Shimadzu GC-14A gas chromatograph equipped with a flame ionization detector and a fused silica capillary column SupelcowaxTM10	Human frontal cortex	Decreased levels in AD patients	[31]
LA, AA, ALA, EPA, and OA	HILIC-MS and GC-MS Shimadzu QP-2010 with an AOC-20S autosampler	Human temporal gyrus	Decreased levels in AD patients	[32]
DHA	LC-MS Agilent 1100 liquid chromatograph coupled to a 1946D mass detector equipped with an electrospray ionization interface (Agilent Technologies)	Human Hippocampus and liver	Decreased levels in AD patients and negative correlation with disease severity	[33,34]
12-HETE, 15-HETE, and PGD_2_	RP-LC-ESI-TOFMS Finnigan LTQ linear ion trap mass spectrometer (Thermo Fisher Scientific) interfaced with a Shimadzu Prominence HPLC system (Waters Inc.)	Plasma and brain of APP/tau mice	Increased 12-HETE levels Decreased 15-HETE and PGD_2_ levels	[35]
isoprostane-F_2a_	GC-MS A Micromass Quattro II mass spectrometer (Micromass, Beverly, MA) equipped with a coaxial electrospray source and triple quadrupole analyzer	Human CSF	Increased levels in AD patients and positive correlation with disease severity	[36]
isoprostane-F_2a_	UPLC-MS/MS Waters Acquity UPLC-Xevo TQD system (Milford)	Human plasma	Increased levels in AD	[37,38]
17- HDHA and 15-HETE	UPLC-ToF-MS Acquity UPLC system (Waters, Milford)	Human CSF	Increased levels in AD patients compared to MCI	[39]
TBX_2_ and PGD_2_	GC-MS Finnigan 4500 GC-MS with the Super Incos Data System (Finnigan) and a capillary column of SPB-1	Human cortex	Increased levels in AD patients	[40,41]
PGE_2_ and 6-keto-PGF_1_	GC-MS Instrument not specified	Human CSF	Increased PGE_2_ and decreased 6-keto-PGF_1_ levels in AD patients	[42]
PGE_2_	LC-MS Instrument not specified	Human CSF	Decreased PGE_2_ levels along progressive learning impairment in AD patients	[43]
F_2_-isoprostanes, PGF_2a_, 8-isoPGF_2a_, and 11-dehydro-TXB_2_	GC–MS Aglient GC–MS system, which consists of an HP 6890 GC and an HP 5973 MSD	Human urine	Increased levels in AD patients	[44]
NPD1	LC-PDA-ESI-MS/MS TSQ Quantum (Thermo Electron Corp.) triple quadrupole mass spectrometer and electrospray ionization	Human hippocampal CA1 region	Reduced in AD patients	[45]
NPD1	LC-PDA-ESI-MS/MS TSQ Quantum (Thermo-Finnigan) triple quadrupole mass spectrometer and electrospray ionization	Hippocampus of 3xTg-AD mice	Reduced in 12–13-month-old mice compared to 4-month-old mice	[46]
LXA_4_ and MaR1	LC-MS-MS Qtrap 5500 equipped with a Shimadzu LC-20AD	Hippocampus of AD patients	Reduced in AD patients compared to controls	[47]
PD1, MaR1, and RvD5	LC-MS/MS LC-20AD HPLC and a SIL-20AC auto-injector paired with a Qtrap 6500	Entorhinal cortex of AD patients	Decreased in AD patients compared to controls	[48]
15-R-LXA_4_	LC-MS/MS Agilent 6470 Triple Quad LC-MS/MS system coupled to an Agilent 1290 HPLC system	Neurons from APP/PS1, APP/PS1/SphK1 and WT mice	Decreased in APP/PS1 mice compared to the other two groups	[49]
RvD4, RvD1, PD1, MaR1, and RvE4	LC-MS/MS Xevo TQ-S equipped with Acquity I Class UPLC	CSF of AD, MCI, and SCI subjects	Decreased in AD and/or MCI compared to SCI	[50]
2-AG	LC-APCI-MS Shimadzu HPLC apparatus LC-10ADVP, coupled to a Shimadzu LCMS-2010, quadrupole MS via a Shimadzu APCI interface	Hippocampus of rodents (mice and rats) treated with Aβ	Increased in the hemisphere ipsilateral to the injection of Aβ, 12 days after treatment	[51]
2-AG	LC-MS/MS UPLC system (Waters Inc.) coupled with a triple quadrupole Quattro Premiere/XE mass spectrometer	Brain tissue of APP/PS1 transgenic mouse model after NO_2_ exposition in presence or absence of MAGL inhibitor JZL184	Increased in presence of JZL184	[52]
2-AG	LC-MS/MS Shimadzu HPLC apparatus LC10ADVP coupled to a quadrupole MS Shimadzu LCMS-2010	Blood of AD patients	Increased in AD patients compared to controls	[53]
AEA	LC-ESI-MS 1100-LC system (Agilent) coupled to a 1946D-MS detector equipped with an electrospray ionization (ESI) interface	Midfrontal and temporal cortex post-mortem tissues of AD patients	Decreased in AD patients compared to controls	[54]
2-AG and AEA	LC-MS/MS 6430 triple quadrupole mass spectrometer (Agilent)	Brain tissue from PS1/APP AD mice	Increased compared to their wild-type littermates	[55]
2-AG	LC-ESI-MS 1100 LC-MSD, SL mode (Agilent)	Brain tissue of 5xFAD mice	Increased after administration of MAGL inhibitor JZL184	[56]
AEA, 2-AG, PEA, and OEA	LC-MS/MS Applied Biosystems MDS SCIEX 4000 Q-Trap hybrid triple quadrupole–linear ion trap mass spectrometer model 1004229-F in conjunction with a Shimadzu series 10AD VP LC system	Frontal cortex, hippocampus, and striatum of AβPPswe/PS1ΔE9	Increased AEA and OEA levels in the areas of both AβPPswe/PS1ΔE9 and wild-type mice with age. Increased AEA levels in AβPPswe/PS1ΔE9 compared to wild-type mice Lower 2-AG levels in AβPPswe/PS1ΔE9 compared to wild-type mice	[57]

**Table 2 ijms-23-06235-t002:** Bioactive lipids evaluation in Parkinson’s disease by lipidomics.

Bioactive Lipid	Methods of Detection/Instruments	Tissue	Variation	References
GlcCer and SM	LC/MS/MS Triple quadrupole mass spectrometer (AB Sciex API 500).	Human CSF	Increased GlcCer levels and decreased SM levels in early stages of de novo PD patients	[93]
Cer (16:0, 18:0, 20:0, 22:0, and 24:1), MonohexosylCer (16:0, 20:0, and 24:0), and LactosylCer	HPLC coupled to/ESI/MS/MS Triple quadrupole mass spectrometer (AB Sciex API3000s)	Human plasma	Increased in PD patients with cognitive impairment	[94]
Cer HexosylCer and SM	Shotgun lipidomics Q Exactive mass spectrometer (Thermo Scientific) equipped with a TriVersa NanoMate ion source (Advion Biosciences)	L444PGBA-mutated human fibroblasts	Increased Ceramide and SM levels and decreased total phospholipid levels in L444PGBA fibroblasts compared to healthy controls and idiopathic PD cells	[95]
Cer	LC-MS Shimadzu High Performance LC system (CBM-20 A, equipped with the binary pump LC-20AB)	Brain of LRRK2^−/−^ mice	Increased in LRRK^−/−^ mice compared to wild-type mice	[96]
Cer, PE, and SM	UHPLC/Q-TOF-MS Dionex UltiMate 3000 UHPLC system (ThermoFisher Scientific) coupled to an ultra-high resolution Maxis II Quadrupole Time-of-Flight (QtoF) mass spectrometer equipped with an electrospray ionization source	Human blood serum	Increased in PD patients	[97]
Long-chain Fas (14:0, 17:1, and 20:1)	UPLC-MS/MS UPLC-ESI-Q-TOF (Agilent) high resolution mass spectrometer	Human plasma	Increased at baseline and decreased in the follow-up in PD	[98]
LPC (16:0 and 18:1) PC (24:0, 24:1, 26:0, 28:0, 31:5, 34:2, 36:4, 36:5, 38:6, 40:5, 42:4, and 44:5)	HPLC-ESI-MS/MS 4000 QTRAP ESI-MS/MS Hybrid Triple Quadrupole/Linear Ion Trap (AB Sciex)	Substantia nigra of 6-OHDA rats	Increased LPC levels in 6-OHDA rats Decreased PC and LC levels in 6-OHDA rats	[99]
BMP 42:8 and Pl 42:10 PC 36:3, PE 36:2, PS 36:3, and SM (18:1/14:0,18:1/16:0)	UPLC-MS Fusion mass spectrometer (Thermo Scientific)	Human substantia nigra	Increased BMP and PI in PD patients Decreased PC, PE, PS, and SM in PD patients	[100]
Cer (16:0 and 18:0) and Hydroxyceramide (18:0)	ESI-HR-MS Orbitrap Thermo Q Exactive mass spectrometer (Thermo Scientific)	Human putamen	Decreased in PD patients	[101]
SM	ESI-MS Hybrid triple quadrupole ion trap mass spectrometer (QTRAP 5500; AB SCIEX) coupled to an attached chip-based automated nanospray source (Triversa NanomateVR)	Human anterior cingulate cortex	Decreased in PD patients	[102]
SM	HP-TLC Silica gel HP-TLC plates (Merck) using a Camag Linomat V semiautomatic TLC spotter (Camag Scientific Inc.)	Human substantia nigra	Decreased in PD patients	[103]
SM (18:1/22:1, 18:1/22/0, 18:1/24:1, 18:1/24:0, and 18:0/24:0) Cer (18:0/18:0, 18:1/24:1, and 18:0/24:1)	HPLC/MS HPLC 1200 system (Agilent) coupled with an Applied Biosystem Triple Quadrupole/Ion Trap mass spectrometer (3200 Qtrap)	Human visual cortex	Increased SM and Cer levels in PD patients	[104]
TAG, SAFA, MUFA, PC, Cer, and SM	UPLC-ESI-qToF-MS/MS Acquity-LCT Premier XE system coupled to an Acquity-Xevo G2QTOF (Waters Corp.)	Human CSF	Increased in PD patients	[105]
TAG (50:5) and Cer (42:0, 40:0, 38:1)	UHPLC-qToF-MS/MS Ultimate 3000 UHPLC (Thermo Scientific) coupled to a Synapt G2-Si QtoF mass spectrometer (Waters)	Human sebum	Decreased in drug-naive and medicated PD sebum samples compared to healthy controls.	[106]
PI (34:1), PS (36:1), and LPC (16:0 and 18:1)	MALDI-TOF/MS) and TLC Bruker Microflex LRF mass spectrometer and Bruker Daltonics Ultraflex Extreme MALDI/TOF mass spectrometer (Bruker Daltonics)	Human parkin-mutant fibroblasts	Increased in parkin^−/−^ PD fibroblasts	[107]
PUFAs (18:3, 20:4, 22:4, 22:5, and 22:6)	GC/MS Hewlett Packard 6890 gas chromatograph using a mass selective detector (HP 5973) equipped with an MS Chemstation	Brain of α-syn^−/−^ mice	Decreased in α-syn^−/−^ mice compared to wild-type littermates	[108]
DHA (22:6) and AA (20:4)	TLC Shimadzu GC-14A gas chromatograph	Lipid rafts from human frontal cortex	Decreased in PD patients	[109]
AA, 13-hydroxy-octadecatrienoic DHA, lyso-PFA, 12-hydroxy-eicosatetraenoic acid, dihydroxy-eicosatrienoic acids, dihidroxy-octadecenoic acids, 17,18-dihydroxy-eicosatetraenoic acid, and hydroperoxy-octadecadienoic acids	UPLS-MS 8045 series UPLS-MS (Shimadzu) coupled with a C8 column (Phenomenex)	Human plasma	Increased AA and 13-hydroxy-octadecatrienoic in PD patients compared to healthy controls Decreased DHA, lyso-PFA, 12-hydroxy-eicosatetraenoic acid, dihydroxy-eicosatrienoic acids, dihidroxy-octadecenoic acids, 17,18-dihydroxy-eicosatetraenoic acid, and hydroperoxy-octadecadienoic acids in PD patients compared to healthy controls	[110]
LTB_3_ and Lyso-PC (18:2)	UPLC-qTOF-MS Agilent 1290 UPLC system coupled with an Agilent 6520 TOF-MS analyzer	Human plasma	Increased levels in plasma of PD patients	[111]
PGE_2_, PGD_2,_ and PGF_2α_	RF-LC-ESI-MS/MS Quadrupole mass spectrometer (API3000, Applied Biosystem) equipped with a TurboIonSpray ionization source	Brains of α-syn^−/−^ mice	Increased levels in α-syn^−/−^ mice compared to wild-type littermates	[112]
PGB_1_, PGH_2_, and 15(*S*)-HETE	ESI-FT-ICR-MS Fourier transform ion cyclotron resonance mass spectrometer (ICR-FTMS, Solarix) equipped with a 12 T superconducting magnet (Magnex Scientific, Varian Inc.) and an electrospray source (Bruker Daltonics)	Brains of manganese-supplemented rats	Increased levels compared to standard diet-fed rats	[113]
F_2_-IsoPs, F_4_-NPs, and HETEs	GC-MS Hewlett Packard 6890 Gas chromatographer coupled to a Hewlett Packard 5973N mass selective detector (Agilent)	Human plasma	Increased levels in early-stage PD patients	[114]
Oxidizable PUFAs containing cardiolipin	LC/MS Dionex Ultimate™ 3000 HPLC coupled to a linear ion trap mass spectrometer (LXQ, ThermoFisher Scientific)	Substantia nigra and plasma of rotenone-lesioned rats	Decreased levels compared to control rats	[115]
AEA	LC-ESI-MS/MS Hybrid triple quadrupole-ion trap mass spectrometer QTRAP 5500 or 6500+ (Sciex) equipped with a Turbo-V-source operating in positive ESI mode	Human plasma	Reduced levels in PD patients	[116]
AEA and 2-AG	LC-ESI-MS/MS HP-MS 5989B quadrupole mass analyzer equipped with an electron impact source	Globus pallidus and substantia nigra of reserpine-treated rats	Increased 2AG levels in the globus pallidus	[117]
AEA	GC-EI-MS VG Micromass model QUATTRO spectrometer	Striatum of 6-OHDA-lesioned rats	Increased AEA levels in the striatum	[118,119]
AEA and 2-AG	HPLC/MS HP 1100 Series HPLC/MS system equipped with a Hewlett Packard octadecyl-silica (ODS) Hypersil column	Caudate–putamen, globus pallidus, and substantia nigra of 6-OHDA-lesioned rats	Reduced AEA levels in the caudate–putamen ipsilateral to the lesion	[120]
AEA and 2-AG	LC–MS LC-Q/TOF-MS System (1290 Infinity LC, 6530 UHD and Accurate-Mass Q-TOF/MS, Agilent)	Striatum from 6-OHDA lesioned rats	Decreased AEA and 2-AG levels after chronic L-DOPA administration	[121]
AEA and 2-AG	LC-APCI-MS Shimadzu (LCMS-2010) quadrupole MS via a Shimadzu APCI interface	Basal ganglia (striatum, globus pallidum, and substantia nigra) of MPTP-lesioned cynomolgus monkeys	Increased AEA and 2-AG levels in the striatum Increased AEA levels in the external globus pallidus Increased 2-AG levels in the substantia nigra	[122]
2-AG	LC-MS Agilent G6410B QQQ instrument	Brain of *Mgll*^+/+^, ^+/−^, and ^−/−^ MPTP-lesioned mice	Increased levels compared to wild-type littermates	[123]
2-AG	LC–MS/MS Triple quadrupole mass spectrometer (Thermo Surveyor PDA/TSQ Quantum, Thermo Scientific)	Ventral midbrain of MPTP-lesioned mice	Increased levels compared to control mice	[124]
AEA, 2-AG, PEA, and OEA	LC-MS/MS 1100 HPLC system coupled to a triple quadrupole 6460 mass spectrometer (Agilent)	Striatum of 6-OHDA, LPS, rotenone, or Poly(I:C)-lesioned rats	Increased AEA and 2-AG levels upon LPS treatment Increased AEA upon rotenone treatment Increased 2-AG upon Poly(I:C) treatment Increased PEA and OEA upon 6-OHDA, LPS, rotenone, and Poly(I:C) treatment	[125,126]
AEA, 2-AG, PEA, and OEA	LC-MS/MS Agilent 1260 infinity 2 HPLC system coupled to a triple quadrupole SCIEX QTRAP 4500 mass spectrometer	Striatum and substantia nigra of AAV-GFP- or AAV-α-syn-lesioned rats	Reduced 2-AG levels at 12 weeks compared to control rats	[127]
AEA	RP-HPLC and HPLC-LIF PerkinElmer Nelson Model 1022	Human CSF	Increased levels in PD patients	[128,129]
AEA and 2-AG	UHPLC-MS/MS and DPX-UHPLC-MS/MS Two dimensional UHPLC–MS/MS system coupled with a Xevo^®^ TQ-D triple-quadrupole operating in positive electrospray ionization	Human plasma and CSF	Increased AEA levels in the CSF of PD patients Decreased 2-AG levels in the plasma and CSF of PD patients	[130,131]
AEA and 2-AG	In-tube SPME-MS/MS Two-dimensional UHPLC–MS/MS system coupled with a Xevo^®^ TQ-D triple-quadrupole operating in positive electrospray ionization	Striatum of 6-OHDA-lesioned rats	Increased AEA and decreased 2-AG in the striatum tissue ipsilateral to the lesion compared to contralateral to the lesion	[132]

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
