# Peer review of "Lipidomics of Bioactive Lipids in Alzheimer’s and Parkinson’s Diseases: Where Are We?"

_ijms, 2022, doi:10.3390/ijms23116235_

Round 1

Reviewer 1 Report

The article is written on a current topic. At the same time, quite a lot of modern and topical review articles are devoted to the role of biologically active lipids in the development of AD and PD. Some of them describe the state of the problem for each of the diseases in much more detail, considering more than 500 references. However, the submitted manuscript may be of interest as a more concise and focused work. At the same time, the article has the following suggestions, which are aimed at improving its accessibility for readers:

1. Add an abbreviation section.
2. Since a large amount of information is given, for better structuring and perception by the reader, it is recommended to make at least two tables with the role of bioactive lipids - for AD and for PD. Perhaps the authors would prefer the resulting tables for each of the families of bioactive lipids for each of the diseases, or the corresponding schemes, which are also highly desirable to be done with references.
3. Despite the fact that the methods of lipidomic analysis are mentioned in the article, lipidomics itself is lost behind the content and role of individual substances in the pathogenesis of diseases in the present version of the manuscript. To match the title of the work, it is recommended to highlight in each of the sections the applied analytical methods, their advantages and limitations.
4. Lines 217-224 are recommended to be rephrased. Since the role of AA in the pathogenesis of AD remains unclear from this paragraph.
5. Line 333 "the" - something is missing.
6. Lines 371, 586 - missing points.
7. Lines 482-486 are redundant italics.

Author Response

The article is written on a current topic. At the same time, quite a lot of modern and topical review articles are devoted to the role of biologically active lipids in the development of AD and PD. Some of them describe the state of the problem for each of the diseases in much more detail, considering more than 500 references. However, the submitted manuscript may be of interest as a more concise and focused work.

R: We thank the reviewer for the kind appreciation of our article. We want to point out that although there are many reviews on this topic, this is the first one reporting only the detections/changes of bioactive lipids by lipidomics and mass spectrometry methodology, which is the golden standard for lipid detection, but without focusing on the effects of these lipids or on their receptors or mechanisms. Our aim, as suggested by the title, is only to review the detection of lipids by such methodology and will help further researchers to better identify which lipids have still to be measured in humans or animals and in which tissues.

At the same time, the article has the following suggestions, which are aimed at improving its accessibility for readers:

  1. Add an abbreviation section. R: Done, as suggested.
  2. Since a large amount of information is given, for better structuring and perception by the reader, it is recommended to make at least two tables with the role of bioactive lipids - for AD and for PD. Perhaps the authors would prefer the resulting tables for each of the families of bioactive lipids for each of the diseases, or the corresponding schemes, which are also highly desirable to be done with references. R: As wisely suggested, we added a figure and two tables to strengthen the manuscript and to make it more attractive for the readers.
    3. Despite the fact that the methods of lipidomic analysis are mentioned in the article, lipidomics itself is lost behind the content and role of individual substances in the pathogenesis of diseases in the present version of the manuscript. To match the title of the work, it is recommended to highlight in each of the sections the applied analytical methods, their advantages and limitations. R: We thank the Reviewer for this suggestion. In the two tables that we prepared, each lipidomic analytical method was detailed for every bioactive lipid, including the instruments.
    4. Lines 217-224 are recommended to be rephrased. Since the role of AA in the pathogenesis of AD remains unclear from this paragraph. R: Corrected, as suggested.
    5. Line 333 "the" - something is missing. R: Corrected, as suggested.
    6. Lines 371, 586 - missing points. R: Corrected, as suggested.
    7. Lines 482-486 are redundant italics. R: Corrected, as suggested.

Reviewer 2 Report

The article "Lipidomics of bioactive lipids in Alzheimer’s and Parkinson’s 2 diseases: where are we at?" is very interesting topic and the authors described the role of lipids in AD and Parkinson's very beautifully step by step. There are also many articles same to that data published but this article contains a lot of new data. I would recommend some changes as I think that these changes will make the paper more attractive for readers.

  1. I recommend authors to at least draw some figures for the article as the figures are the backbone of the article beauty. the authors can follow these articles. https://www.mdpi.com/1422-0067/21/4/1505   https://www.mdpi.com/2073-4409/8/1/27  
  2. The authors can also add some mechanisms diagrams of lipids' role in AD and Parkinson's and can add at least 1-2 tables in the article. 

Author Response

The article "Lipidomics of bioactive lipids in Alzheimer’s and Parkinson’s 2 diseases: where are we at?" is very interesting topic and the authors described the role of lipids in AD and Parkinson's very beautifully step by step. There are also many articles same to that data published but this article contains a lot of new data. I would recommend some changes as I think that these changes will make the paper more attractive for readers.

  1. I recommend authors to at least draw some figures for the article as the figures are the backbone of the article beauty. the authors can follow these articles. https://www.mdpi.com/1422-0067/21/4/1505  https://www.mdpi.com/2073-4409/8/1/27  
  2. The authors can also add some mechanisms diagrams of lipids' role in AD and Parkinson's and can add at least 1-2 tables in the article.

R: We thank the reviewer for the kind appreciation of our article. We want to point out that although there are many reviews on this topic, this is the first one reporting only the detections/changes of bioactive lipids by lipidomics and mass spectrometry methodology, which is the golden standard for lipid detection. As wisely suggested, we added a figure and two tables to strengthen the manuscript and to make it more attractive for the readers.